# Assessments of Water-Soluble Inorganic Ions and Heavy Metals in Atmospheric Dustfall and Topsoil in Lanzhou, China

**DOI:** 10.3390/ijerph17082970

**Published:** 2020-04-24

**Authors:** Yingquan Li, Baowei Zhao, Kaixiang Duan, Juexian Cai, Wujiang Niu, Xiao Dong

**Affiliations:** 1School of Environmental and Municipal Engineering, Lanzhou Jiaotong University, Lanzhou 730070, China; yingquanli@mail.lzjtu.cn (Y.L.); 022412125@hncj.edu.cn (K.D.); caijuexian@sina.com (J.C.); 2Gansu Dust Suppression for Transportation and Storage Engineering Research Center, Lanzhou 730070, China; xiaodong_lztj@sina.com; 3Lanzhou Tianji Environmental Protection Limited Company, Lanzhou 730070, China; 4Environmental Monitoring Center Station in Gansu Province, Lanzhou 730020, China; wujiangniu1958@sina.com

**Keywords:** dustfall, topsoil, water-soluble inorganic ions (WSIIs), heavy metals, contamination characteristics, Lanzhou

## Abstract

The chemical features of atmospheric dustfall and topsoil in the same region could reflect the processes of the migration, transport, and diffusion of pollutants in the atmospheric-soil system. Samples of atmospheric dustfall and topsoil were collected in Lanzhou City. The contents and correlation of water-soluble inorganic ions (WSIIs) and heavy metals in dustfall and topsoil were analyzed, the sources of heavy metals and WSIIs in dustfall were distinguished, and the potential ecological risks of heavy metals in dustfall and topsoil were evaluated. The highest contents of WSIIs are SO_4_^2−^ (18,594 mg·kg^−1^) and Ca^2+^ (10,070 mg·kg^−1^) in dustfall, and for SO_4_^2−^ (8271 mg·kg^−1^) and Na^+^ (1994 mg·kg^−1^) in topsoil. The concentrations of heavy metals (Pb, Cu, Zn, Cr, Cd, and Ni) in dustfall are considerably higher than those in topsoil. Combustion of biomass and coal, transportation and industrial activities are the major anthropogenic sources of WSIIs and heavy metals in Lanzhou. Pollution of heavy metals except Cr and Ni in dustfall, and Cu, Cr, and Ni in topsoil was up to different degrees, where the pollution of Cd was serious. The risk of Cd in dustfall is high while moderate in topsoil. This research could offer a reference for the atmospheric particle pollution prevention and control in Lanzhou.

## 1. Introduction

Atmospheric dustfall is one of the main urban atmospheric particulate pollutants [1,2], and it has important environmental indication function [3]. As an indirect carrier, atmospheric dustfall brings pollutants from the atmosphere to the land surface or water, seriously affecting aquatic and terrestrial ecosystem [4]. With the continuous economic development and the urbanized advancement acceleration, the amounts of atmospheric dustfall and its carrying pollutants are increasing. As an important part of atmospheric particles, water-soluble inorganic ions (WSIIs) generally account for 20~60% of the mass concentration of particles [5,6], and even up to 80% [7]. WSIIs such as sulphate, nitrate, and ammonium are hygroscopic [6], which affect the visibility of atmosphere through the absorption and scattering of light [8]. They can also affect the acidity and alkalinity of atmospheric precipitation [9]. In addition, the atmospheric dustfall’s contribution rate to the accumulation of heavy metals in topsoil ranks first among various exogenous input factors [10]. Heavy metal pollutants are not to be degraded and will cause long-term damage to human health and ecological environment [11,12]. Therefore, the study on the contamination features of WSIIs and heavy metals in the atmospheric dustfall and topsoil has become a research focus in environmental science field in recent years.

In 1975, Day et al. [13] first proposed the new term “city street dust fall.” Since then, scholars have studied it in various aspects, including particle size characteristics [14], temporal distribution [15], spatial distribution [16], and chemical composition [17,18]. At present, the researches on WSIIs in atmospheric particles mainly focus on the analysis of pollution characteristics [6], source analysis [7], transformation mechanism between gaseous pollutants and WSIIs [19], and the effects on aerosol optical properties [8]. The researches on heavy metals in atmospheric particulates mainly focus on the heavy metal content [20], spatial distribution [21], pollution characteristics, source identification, ecological risk, and health risk assessment [22]. Currently, many studies on air particle pollution in China have paid close attention to the chemical composition, occurrence form, and source analysis. However, the research areas of both WSIIs and heavy metals in atmospheric particles are mostly concentrated in large and medium-sized cities or developed regions such as Beijing-Tianjin-Hebei region [23,24,25] and Yangtze River Delta [26]. There is very little research on atmospheric dustfall in arid and underdeveloped areas in the northwest China. Besides, analysis combining the data from heavy metals with WSIIs could provide interesting information, but there are few studies in this field.

Lanzhou is the capital of Gansu Province. It is located in the west of Loess Plateau and the Yellow River valley basin in the northeastern of Qinghai-Tibet Plateau, which is a typical “valley belt” city. The soil texture of Lanzhou is IV class self-weight collapsible loess, which is easy to produce dust pollution after human disturbance. The climate in this area is arid and the temperature inversion is serious in winter. The external input of sand dust from the surrounding deserts, a lot of smoke and dust emissions from industrial enterprises, exhausts from vehicles, and the continuous acceleration of construction results in a prominent PM contamination in Lanzhou City. Because of the special topography and climate, the atmospheric PM contamination pattern in Lanzhou may be different from those in other cities which are located in plain areas, for example, Beijing-Tianjin-Hebei region and Yangtze River Delta. Therefore, studying WSIIs and heavy metals in atmospheric dustfall and topsoil will be helpful to understand the regularities of their migration, transport, and diffusion in the atmospheric-soil system.

This article takes the main urban area of Lanzhou as the research area. The contents and correlation of WSIIs (Cl^−^, F^−^, NO_2_^−^, NO_3_^−^, SO_4_^2−^, K^+^, Na^+^, Mg^2+^, and Ca^2+^) and heavy metals (Pb, Cu, Zn, Cr, Cd, and Ni) in dustfall and topsoil were studied. The sources of WSIIs and heavy metals in dustfall were distinguished. The potential ecological risks of heavy metals in dustfall and topsoil were evaluated, with an aim to supply a reference for the atmospheric particle pollution prevention and control in Lanzhou.

## 2. Materials and Methods

### 2.1. Study Area and Sample Collection

Lanzhou is located at the Yellow River Valley and surrounded by hills. In the south there are Gaolan mountain and other loess hills, with an altitude of 1700~2500 m, which are 180~1000 m higher than that of Lanzhou City; in the north there are Loess hills and low or middle mountains, with an altitude of about 1700 m, which are about 180 m higher than that of Lanzhou City. The landform of the main urban area of Lanzhou City is erosion accumulation valley plain. The geological structure of Lanzhou City belongs to the middle uplift zone of Qilian in Kunlun-Qinling geosyncline fold system. Because the Yellow River runs through Lanzhou City, local groundwater is relatively rich. The groundwater in Lanzhou City is mainly loose rock pore phreatic water. Landslides, collapses, and debris flows are the main types of geological disasters in Lanzhou City. The average annual temperature is 10.3 °C. The annual average sunshine hours are 2446, the frost-free period is 180 days, and the average annual precipitation is 327 mm, which mainly concentrate from June to September. It is an important industrial base and transportation hub in northwest China. The main urban areas of Lanzhou (Chengguan District, Anning District, Qilihe District, and Xigu District, excluding suburban counties) were selected as the research areas. The atmospheric dustfall and topsoil samples were obtained at each sample position. The sampling sites are shown in Figure 1.

The samples were collected in July 2017, and there was no rainfall during the week before the collection. The atmospheric dustfall was sampled at a height of 1.0 to 2.0 m (being close to the height of human respiratory tract exposure) using hairbrush to collect the dust on clean paper and transfer it into self-sealing bag. Sampling was kept away from the demolition, factories, and other areas that could cause local pollution. Four samples were collected at each position by the three-point mixed method, which means that three samples are collected at each sample point and mixed as one atmospheric dustfall sample. A total of 40 samples were obtained. Topsoil samples (depth 2–10 cm) were collected on the bare ground or green belt near the sampling points of atmospheric dustfall by the three-point mixed method, and a total of ten samples were obtained. Both the atmospheric dustfall samples and the topsoil samples were pretreated as follows: the branches, leaves, etc., in the sample were removed. The samples were laid in a beaker and dried in an oven (DZF—6020A, Shanghai Lichen Instrument Technology Co., Ltd., Shanghai, China) at 105 °C for 3 h, and then the samples were crushed and passed through 20 mesh and 200 mesh sieves for reserve. The quartering method was applied for sample weighting.

### 2.2. Materials

The chemicals NaF, NaCl, KNO_3_, NaNO_2_, NaNO_3_, Na_2_SO_4_, Mg(NO_3_)_2_·6H_2_O, and Ca(NO_3_)_2_·4H_2_O with superior grade purity, standard reserve solutions (1000 mg·L^−1^) containing Cl^−^, F^−^, NO_2_^−^, NO_3_^−^, SO_4_^2−^, Na^+^, K^+^, Mg^2+^, and Ca^2+^, high purity argon (purity ≥ 99.999%), standard reserve solutions of Cu, Zn, Cr, Cd, Ni, Pb (1000 mg·L^−1^), internal standard solutions of Rh (10 μg·mL^−^^1^), mass spectrometry tuning solution (Be, In, U, etc. 10 mg·L^−1^), and the reagents HNO_3_, HF, HCl, CH_4_N_2_S, and H_2_O_2_ with superior grade purity were bought from Baiyin Liangyou Chemical Reagents CO., LTD (Baiying, China). Soil for extractable trace elements (GBW07437, National Institute of Metrology, Beijing, China). Quality Control Materials (QCM-TW001 and QCM-CW001, Institute for Environmental Reference Materials of Ministry of Environmental Protection, Beijing, China). All standard solutions were prepared with ultrapure water (CSR-1-10, > 18.2 MΩ·cm, Beijing ASTK Technology Development Co., Ltd., Beijing, China).

### 2.3. Analytical Methods

For ion determination, 0.1 g of sample was weighed placed into a 500-mL centrifugal tube and 100 mL of ultrapure water was added. The tube cap was tightened, and ultrasonic extraction in water bath was conducted at 25 °C for 30 min. The tube was rotated at 4000 r·min^−1^ for 15 min. The supernatant was taken and filtered through a 0.22-μm membrane. All the samples were analyzed parallelly three runs with blank. Instead of weighing the sample, blank sample was prepared according to the same steps. WSIIs (Cl^−^, F^−^, NO_2_^−^, NO_3_^−^, SO_4_^2−^, K^+^, Na^+^, Mg^2+^, and Ca^2+^) were determined by ion chromatograph (881 Compact IC pro, Metrohm, Herisau, Switzerland)). The detection limits of Cl^−^, F^−^, NO_2_^−^, NO_3_^−^, SO_4_^2−^, K^+^, Na^+^, Mg^2+^, and Ca^2+^ were 0.002, 0.001, 0.004, 0.005, 0.004, 0.002, 0.001, 0.001, and 0.003 mg·L^−1^, respectively. The standard curve correlation coefficient was above 0.995. The relative quasi-deviation of the three measurements was less than 9% and the recovery rates ranged from 86.0% to 89.7%. The anion column was Metrosep A Supp 5-150 (Metrohm, Herisau, Switzerland). 3.2 mmol·L^−1^ Na_2_CO_3_ and 1 mmol·L^−1^ NaHCO_3_ were the eluent. The cationic column was Metrosep C4-150 (Metrohm, Herisau, Switzerland), and 5.6 mmol·L^−1^ HNO_3_ was the eluent.

As for heavy metal determination, a pretreatment was carried out for samples with an Automatic Digestion Instrument (Politech DigestLinc ST60D, Beijing, China). Total of 0.5 g of each sample was weighed and put into Teflon beaker, 10 mL of HNO_3_-HCl mixed solution (167.5 mL HCl and 55.5 mL HNO_3_ were added to 500 mL ultrapure water, with a constant volume up to 1 L) was added. The beaker was covered and heated for 2 h at 100 °C, then shaken well and cooled for 20 min, diluted to 50 mL with ultrapure water, stood for 2 h and measured after filtration. Inductively coupled plasma mass spectrometry (ICP-MS X series II, Thermo Fisher Scientific (China) Co., Ltd., Shanghai, China) was used to determine the contents of metal elements. Soil for extractable trace elements (GBW07437) were selected as solid reference materials to draw working curve, which could basically eliminate the matrix interference in the determination of heavy metals in environmental soil by ICP-MS. Quality control materials (QCM-TW001 and QCM-CW001) were selected as reference solution for ICP-MS analysis. The standard curve correlation coefficient was above 0.9980. The relative quasi-deviation of the three measurements was less than 10%. The recovery rates are higher than 90%.

### 2.4. Assessment Methods

Correlation analysis (CA) was used to characterize the variation rule of each element and the correlation degree among measured variables. The communality of each indicator was extracted by principal component analysis (PAC), then the compositional patterns were compared among the samples to find out the mutual influence factors. SPSS 22.0 software (IBM, New York, NY, USA) was used for PCA and CA.

Geo-accumulation index (*I*_geo_) is an index that reflects the effect of man-made activities and natural geological processes on heavy metal pollution [27,28], and it is described by the Equation (1).
(1)$$I_{geo}=log_{2}\left(\frac{C_{i}}{1.5\times B_{i}}\right)$$
where *B_i_* is the environmental background value in Gansu province [29], *C_i_* refers to the measured value of the metal *i.* 1.5 is a correction factor related to the geological and sedimentary characteristics of rocks. If *I*_geo_ ≤ 0, it means the medium is uncontaminated; 0 < *I*_geo_ < 1, from uncontaminated to moderately contaminated; 1 < *I*_geo_ < 2, moderately contaminated; 2 < *I*_geo_ < 3, from moderately to strongly contaminated; 3 < *I*_geo_ < 4, strongly contaminated; 4 < *I*_geo_ < 5, from strongly to extremely contaminated; and *I*_geo_ > 5, extremely contaminated [30].

The degree of the potential environmental risk of heavy metals could reflect by potential ecological risk index (*RI*), which is a comprehensive index that regards environmental effect, ecological effect, and toxicity [31].(2)$$RI=\sum_{i=1}^{n}E_{r}^{i}=\sum_{i=1}^{n}\left(T_{r}^{i}\times \frac{C_{i}}{C_{0}}\right)$$
where *E_r_^i^* means the single factor potential ecological risk index. *C_i_* refers to the measured value of heavy metal *i* in the sample. *C*_0_ means heavy metal *i* environmental background value in Gansu province [29]. *T_r_^i^* is toxic response factor of a given element, which for Zn, Cr, Cu, Ni, Pb, and Cd are 1, 2, 5, 5, 5, and 30, respectively [31]. On the basis of Hakanson’s research, the single factor potential ecological risk is usually regarded as safety while *E_r_^i^* < 40, slight 40 ≤ *E_r_^i^* < 80, moderate 80 ≤ *E_r_^i^* < 160, considerable 160 ≤ *E_r_^i^* < 320, and high *E_r_^i^* ≥ 320. The comprehensive potential ecological risk is usually regarded as low while *RI* < 150, moderate 150 ≤ *RI* < 300, considerable 300 ≤ *RI* < 600, and high *RI* ≥ 600 [31].

## 3. Results and Discussion

### 3.1. The Contents of WSIIs and Heavy Metals in Dustfall and Topsoil

As illustrated in Table 1, the contents of SO_4_^2−^ and Ca^2+^ in dustfall are higher than those of other WSIIs, with an average content of 18,594 and 10,070 mg·kg^−1^, respectively. Ca^2+^ is the main element of construction [32], and SO_4_^2−^ mainly comes from the secondary conversion of SO_2_ emitted from industrial coal and the direct emission of wet desulfurization from thermal power plants [33,34]. Therefore, the high content of Ca^2+^ may be associated with the current subway construction and more construction sites in Lanzhou. The high concentration of SO_4_^2−^ may be associated with burning in centralized heating stations and thermal power plants. The contents of SO_4_^2−^ and Na^+^ in topsoil are higher than n those of other WSIIs, with the average content of 8271 and 1833 mg·kg^−1^, probably because the soil type in the area is predominantly saline-alkaline soil [35]. In both dustfall and topsoil, the main WSIIs are SO_4_^2−^, Na^+^, Ca^2+^, Cl^−^, and NO_3_^−^. The contents of WSIIs in the dustfall are much higher than those in topsoil, which indicates that WSIIs in the dustfall are affected by human activities in addition to the natural sources in the nearby soil.

Furthermore, the ratio of NO_3_^−^ and SO_4_^2−^ in atmospheric particles is usually used to judge whether the city air is mainly polluted by flowing sources (such as automobile exhaust) or by stationary sources (such as coal burning) [36]. Ratios being greater than 1 and less than 1 represent flowing and stationary sources [36], respectively. The ratio of NO_3_^−^ and SO_4_^2−^ in this study is 0.09, indicating that atmospheric dustfall in Lanzhou is mainly from stationary sources.

The concentrations of heavy metals in atmospheric dustfall and topsoil are illustrated in Table 2. The average concentrations of heavy metals in atmospheric dustfall are in the order of Zn (379.84 mg·kg^−1^) > Pb (133.78 mg·kg^−1^) > Cr (95.61 mg·kg^−1^) > Cu (84.25 mg·kg^−1^) > Ni (43.13 mg·kg^−1^) > Cd (3.89 mg·kg^−1^). The similar results were observed by Li et al. [37]. The heavy metals average contents in topsoil are sorted from large to small as Zn (136.07 mg·kg^−1^) > Cr (85.25 mg·kg^−1^) > Pb (55.99 mg·kg^−1^) > Cu (42.20 mg·kg^−1^) > Ni (37.61 mg·kg^−1^) > Cd (0.55 mg·kg^−1^), and the contents of Zn, Pb, Cu are slightly higher than those reported by Zhao et al. [38]. This may be related to the time accumulation effect of heavy metals in topsoil, and to some extent indicating that the pollution level of heavy metals in topsoil in Lanzhou is basically stable.

### 3.2. Correlation Analysis of WSIIs and Heavy Metals 

Pearson correlation coefficient is a common way to determine the linear relationship between variables, and its value range is −1~1 [39]. The correlation analysis of ions could preliminarily infer the sources of different ions in dustfall and topsoil [23]. Correlation coefficients of WSIIs and heavy metals in dustfall and topsoil are illustrated in Table 3 and Table 4.

As seen from Table 3, the same WSIIs in topsoil and dustfall are positively correlated, and the correlation degree is sorted from large to small as Na^+^ > SO_4_^2−^ > Cl^−^ > F^−^ > Ca^2+^ > NO_2_^−^ > Mg^2+^ > K^+^ > NO_3_^−^, where the high correlation of Na^+^ and SO_4_^2−^ with a correlation coefficient of 0.989 and 0.988 respectively is found (*P* < 0.01). It shows that there is a process of mutual migration and transformation of WSIIs in dustfall and topsoil, in which Na^+^ and SO_4_^2−^ is more obvious. F^−^ and SO_4_^2−^ in dustfull is significantly correlated with Na^+^ and NO_2_^−^ in topsoil respectively (0.966 and 0.991, *P* < 0.01), and Ca^2+^ in dustfull is correlated with Cl^−^ (0.739), K^+^ (0.779), and Mg^2+^ (0.786) in topsoil respectively (*P* < 0.05), which indicates that there is a good associated relationship between these indicators and their sources.

As illustrated in Table 4. The correlation degree of selected heavy metal between topsoil and dustfall is reordered as Cu > Ni > Zn > Cd > Cr > Pb, being positively correlated, where the content of Cu has a high correlation coefficient of 0.943 (*P* < 0.01). It proves that heavy metals enter the topsoil through dustfall rather than the process of natural soil formation [40], in which it is more obvious for Cu. Cr in dustfull is correlated with Ni in topsoil (0.823, *P* < 0.05), indicating they might be from a common source because of their good associated relationship.

### 3.3. The Sources of WSIIs and Heavy Metals

Normally, it is difficult to distinguish sources precisely based on the results of CA, which means that further analysis needs to be carried out in combination with other methods. Principal component analysis (PCA) was implemented with normalized data. The suitability of the measured data was examined by Kaiser-Meyer-Olkin (KMO) test and Bartlett’s test. The value of KMO test being larger than 0.5 indicates that PCA is effective [41]. In this work, the KMO and Bartlett’s test value of WSIIs is 0.528 and 0 respectively, and the value of heavy metals is 0.560 and 0 respectively, which shows the PCA is valid [42]. A standard scoring function was exploited to distribute 0.1 to 1.0 points for each indicator [43]. The weights of every indicator is equal to the quotient of its communality and the sum of all indicators’ communality [43]. Combining data from WSIIs and heavy metals in the same PCA analysis cloud provide important information which is missing when run the analysis separately. The content of WSIIs and heavy metals were extracted with three principal components (PC1, PC2, and PC3) based on the eigenvalue greater than 1. After the maximum orthogonal rotation of variance, a total of three factors are obtained (RPC1, RPC2, and RPC3), and the cumulative variance contribution rate are 88.64% (Table 5).

As seen from Table 5, RPC1 has a high loading of F^−^, Cu, Pb, Zn, Ni, and Cr, and account for 39.93% of the total variance. RPC1 could be related to transportation and industrial production. It is reported that the fluorides that cause pollution mainly come from the processing industries such as bricks and tiles, cement, ceramics, electrolytic aluminum, and fluorine-containing drugs [44]. Because of the special geological characteristics of Lanzhou (more loess and limestone), there are more brick manufacturing industries in Lanzhou. Clay will escape calcium fluoride, silicon tetrafluoride, and hydrogen fluoride into the atmosphere during high-temperature sintering. In addition, there are aluminum manufacturing enterprises in Lanzhou. A large amount of gas such as hydrofluoric acid and silicon tetrafluoride and dust such as aluminum fluoride and calcium fluoride are generated during the electrolysis process. Cr mainly comes from soil, coal burning, and metal smelting [45]. However, there are few metal smelting enterprises and basically no garbage incineration in Lanzhou. Therefore, Cr is mainly related to coal burning in centralized heating stations and thermal power plants. The usual origins of Pb are industrial fumes, vehicle exhausts, sewage sludge, and lead arsenate pesticides [46]. However, in urban areas, pesticides are less used and sewage sludge is treated uniformly. Although the application of unleaded gasoline has reduced the content of Pb in per unit volume vehicle exhaust, it is believed that traffic source is still a considerable source of Pb in atmospheric environment of Lanzhou city due to the sharp increase of vehicles and the cumulative effect of Pb emissions from traffic sources in history. Some studies indicate that Pb extracted from vehicle exhausts does not diffuse significantly beyond 30 m from the road [47]. Coal burning exhausts, industrial fumes, and lead aerosols could be transferred long distances [48]. Atmospheric deposition is also a major source of Pb. In addition, Zn and Cu are also commonly made into industrial products such as automobile parts [49]. Accordingly, Zn and Cu may come from transportation and industrial pollution. There are many Ni mines in Jinchang City, in the northwest of Lanzhou city, and the migration of aerosol is one of the factors for the existence of Ni in the study area.

RPC2 accounting for 32.39% of variables, are mainly related to WSIIs and affected by Cl^−^, NO_3_^−^, NO_2_^−^, K^+^, Mg^2+^, and Ca^2+^, which may be attributed to combustion and construction dust. K^+^ and Cl^−^ is associated with biomass combustion [50,51], Ca^2+^ and Mg^2+^ is used as an indicator of municipal construction [32], and NO_3_^−^ and NO_2_^−^ is mainly formed by the oxidation of nitrogen oxides produced by combustion [52]. According to RPC1, the sources of WSIIs in the atmospheric dustfall in Lanzhou are extensive and belong to compound pollution. Zhang et al. [53] analyzed and compared the aerosol samples from the same sampling point in winter of 1990 and 2007 in Lanzhou City, and found that the pollution type of WSIIs in Lanzhou City was “sulfuric acid calcium” type in 1990; the pollution type of air aerosol water soluble ions in Lanzhou City changed to multi-element direction in 2007. Zhao et al. [54] collected aerosol samples in Beijing from 2009 to 2010, analyzed the pollution characteristics of water-soluble ions in aerosols, and compared them with the relevant research results in Beijing several years ago. They found that the concentration of water-soluble secondary ions in PM_2.5_ in Beijing increased significantly, and the ratio of NO_3_^−^/SO_4_^2−^ increased significantly compared with that a few years ago, indicating that the pollution in Beijing has completely transformed from coal smoke pollution into a mixture of reactive gases and fine particles dominated by vehicle exhaust. Therefore, because of the development of regional economy, the change of industrial and energy structure and the increase of the number of motor vehicles, the proportion and order of the main WSIIs in atmospheric particulates could change obviously with the passage of time.

RPC3 are responsible for 16.32% of the total variance and should represent natural source because of high loading of SO_4_^2−^ and Na^+^. The soil type in the study area is predominantly saline-alkaline soil, and the soil salt type is mainly Na_2_SO_4_ one [35].

### 3.4. Contribution of Anthropogenic Heavy Metals

The *I*_geo_ values of heavy metals in atmospheric dustfall and topsoil are illustrated in Table 6. The pollution level of Cd in atmospheric dustfall ranges from strong to extreme, Pb from moderately to strongly, Cu and Zn moderately, and Cr and Ni uncontaminated. Cd pollution in the topsoil is moderate, Pb and Zn from uncontaminated to moderate, and Cu, Cr, and Ni uncontaminated. Pb, Zn, and Cu are called urban elements [55]. These three elements are highly enriched in atmospheric dustfall, while less concentrated in topsoil, indicating that Pb, Cu, and Zn elements in dustfall are primarily influenced by artificial activities, and their transport path is from atmosphere to soil. Cd is certain enrichment in both atmospheric dustfall and soil, and its transmission path may be that the atmosphere and soil influence each other.

Liu et al. [56] evaluated the heavy metals *Igeo* in an industrial park of Shaanxi Province in spring, and found that *Igeo* values of heavy metals followed the order of Cd > Pb > Zn > Cu > Cr > Ni. Zhang et al. [57] studied the *Igeo* of various metal elements in the atmospheric dust in Quanzhou City, and found that Ni, Cu, Zn, and Pb pollution is moderate; while Cd serious. It is suggested that Cd pollution is serious in dustfall in China cities, which should be paid more attention. Wang et al. [58] studied the *I*_geo_ of heavy metals in the air dust in the main urban area of Chongqing. They found that Cd is the most serious pollutant of the four heavy metals in dustfall, followed by Pb, Ni, and Cr. The impact of atmospheric dust input on heavy metals in topsoil is not significant.

### 3.5. Potential Ecological Risk

The evaluation values of potential ecological risks of heavy metals in atmospheric dustfall and topsoil are illustrated in Table 7. The single factor ecological risk degree of Cd is high and moderate in the atmospheric dustfall and topsoil, respectively, and those of Pb, Zn, Cu, Cr, and Ni are safe. The comprehensive potential ecological risk degree is high and moderate in the atmospheric dustfall and topsoil, respectively, and is mainly caused by the high single factor ecological risk pollution degree of Cd. In general, the heavy metals in both atmospheric dustfall and topsoil of Lanzhou have certain potential ecological risks. The most seriously affected element is Cd, which should be paid attention to by relevant departments. This is similar to studies in other parts of China, such as Chengdu, Taiyuan, and Tianjin, where the single factor ecological risk index of Cd is 8697, 1203, and 933, respectively. The value of these several is far more than 520, the potential ecological risk is extremely strong [59]. Tian et al. [60] reported the *E_r_^i^* values of heavy metals in atmospheric dustfall in Nanjing were sorted as Cd (370.5) > Cu (41.19) > Pb (23.54) > Zn (10.91) > Ni (8.25) > Cr (1.54). Chen et al. [61] found that the *E_r_^i^* of heavy metals in dustfall in Xi’an is Cd (1334) > Pb (72.5) > Cu (29.3) > Cr (15.7) > Ni (12.4) > Zn (10.4). Zhang et al. [57] observed that the average *E_r_^i^* of elements in dustfall in Quanzhou is Cd (1402) > Pb (32.87) > Cu (27.40) > Ni (24.38) > Zn (11.66) > Cr (6.16). Therefore, Cd has a very high potential ecological risk in atmospheric dustfall of various cities in China. Research by Li et al. [62] in Nanjing shows that atmospheric deposition has an obvious enrichment effect on heavy metals in the topsoil, which is affected by both the heavy metal content and the deposition amount of atmospheric deposition. The accumulation of some heavy metals in soil is closely related to atmospheric deposition.

## 4. Conclusions

WSIIs and heavy metals can have serious impacts on the ecological environment and human health. It is of great significance to study their pollution characteristics in the air–soil ecosystem. The results obtained in this work offer relevant information on the characteristic, sources, and level of WSIIs and heavy metals pollution in atmospheric dustfall and topsoil in Lanzhou. WSIIs have significant correlation with heavy metals. Combustion of biomass and coal, transportation and industrial activities are the major anthropogenic sources of WSIIs and heavy metals in the study area. In the process of pollution prevention, attention should be paid to controlling Cd because of its higher pollution index and potential ecological risks. This study could provide reference for the atmospheric particle pollution prevention and control in Lanzhou. In addition, the results of this study are helpful to understand the topsoil pollution in Lanzhou.

## Figures and Tables

**Figure 1 ijerph-17-02970-f001:**
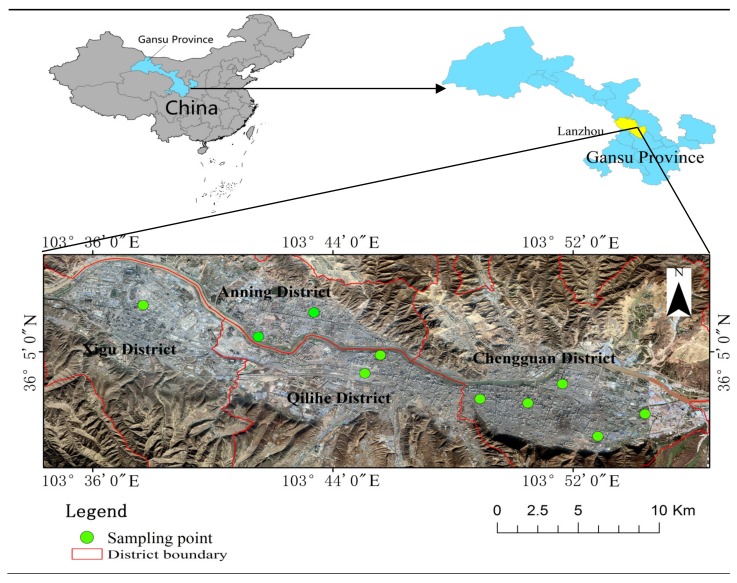
Study area and sampling sites.

**Table 1 ijerph-17-02970-t001:** The contents of water-soluble inorganic ions (WSIIs) in atmospheric dustfall and topsoil (mg·kg^−1^).

Sample	Value	F^−^	Cl^−^	NO_3_^−^	NO_2_^−^	SO_4_^2−^	Na^+^	K^+^	Mg^2+^	Ca^2+^
Dustfall	Min	109	457	192	31	9955	171	1255	19.3	648
	Max	816	3267	3366	397	31,615	21,937	102	812	12,028
	Mean	350	1728	1755	87.6	18,954	3698	668	491	10,070
	SD	195.8	796.1	965.2	43.8	7960.7	1664.1	374.1	176.8	3110
	VC	0.56	0.46	0.55	0.50	0.42	0.45	0.56	0.36	0.31
Topsoil	Min	8.15	29.7	13.3	1.95	60	16.2	42.5	0.708	7.98
	Max	39.8	2471	3375	54.8	31,634	12,202	386	353	9920
	Mean	18.4	747	694	16.2	8271	1994	193	165	1833
	SD	9.57	313.7	381.7	6.97	3391.1	1016.9	94.6	47.9	403.3
	VC	0.52	0.42	0.49	0.43	0.41	0.51	0.49	0.29	0.22

“Min” is the minimum concentration of WSIIs in collected samples; “Max” is the maximum concentration of WSIIs in collected samples; “Mean” is the arithmetic mean concentration of WSIIs in all samples. “SD” is the standard deviation of samples. “VC” is the variation coefficient of samples.

**Table 2 ijerph-17-02970-t002:** The concentrations of heavy metals in atmospheric dustfall and topsoil in Lanzhou (mg·kg^−1^).

Sample	Value	Cu	Pb	Zn	Cr	Cd	Ni
Dustfall	Min	52.57	98.12	302.25	76.54	2.69	32.14
	Max	173.81	216.13	587.21	141.69	5.42	60.39
	Mean	84.25	133.78	379.84	95.61	3.89	43.13
	SD	35.03	33.96	80.44	18.54	0.89	8.74
	VC	0.42	0.29	0.25	0.19	0.23	0.20
Topsoil	Min	25.21	38.24	103.77	57.89	0.34	26.54
	Max	73.36	84.25	219.23	138.25	0.81	39.10
	Mean	42.20	55.99	136.07	85.25	0.55	37.61
	SD	14.79	16.80	32.66	17.91	0.14	7.52
	VC	0.35	0.30	0.24	0.21	0.25	0.12

“Min” is the minimum concentration of heavy metals in collected samples; “Max” is the maximum concentration of heavy metals in collected samples; “Mean” is the arithmetic mean concentration of heavy metals in all samples. “SD” is the standard deviation of samples. “VC” is the variation coefficient of samples.

**Table 3 ijerph-17-02970-t003:** Correlation coefficients of WSIIs in dustfull and topsoil.

	F^−^(d)	Cl^−^(d)	NO_3_^−^(d)	NO_2_^−^(d)	SO_4_^2−^(d)	Na^+^(d)	K^+^(d)	Mg^2+^(d)	Ca^2+^(d)
F^−^(s)	0.837 *	0.241	−0.101	−0.434	0.607	0.663	−0.353	−0.392	−0.230
Cl^−^(s)	−0.160	0.882 *	0.046	−0.413	0.608	−0.223	−0.119	0.250	0.739 *
NO_3_^−^(s)	0.020	0.413	0.105	−0.433	0.744	−0.027	−0.145	0.204	0.652
NO_2_^−^(s)	0.606	0.644	0.187	0.468	0.991 **	0.533	−0.226	−0.018	0.287
SO_4_^2^^−^(s)	0.576	0.568	0.090	−0.546	0.988 **	0.527	−0.325	−0.099	0.239
Na^+^(s)	0.966 **	0.436	0.008	−0.370	0.720	0.989 **	−0.400	−0.475	−0.518
K^+^(s)	−0.128	0.151	−0.042	−0.447	0.525	−0.313	0.122	0.217	0.779 *
Mg^2+^(s)	−0.354	0.169	0.023	−0.332	0.434	−0.403	−0.057	0.313	0.786 *
Ca^2+^(s)	−0.248	0.271	0.089	−0.339	0.534	−0.331	−0.029	0.341	0.824 *

** *p* < 0.01 (2-tailed), * *p* < 0.05 (2-tailed), (d and s represent dustfall and topsoil, respectively).

**Table 4 ijerph-17-02970-t004:** Correlation coefficients of heavy metals in dustfull and topsoil.

	Cu(d)	Pb(d)	Zn(d)	Cr(d)	Cd(d)	Ni(d)
Cu(s)	0.943 **	−0.506	0.213	0.395	−0.273	0.005
Pb(s)	0.201	0.052	−0.103	−0.116	0.252	−0.605
Zn(s)	0.270	−0.248	0.849 *	0.564	−0.188	0.273
Cr(s)	−0.096	−0.822	−0.236	0.369	−0.731	0.254
Cd(s)	−0.528	0.273	−0.720	−0.595	0.725 *	−0.747
Ni(s)	0.125	0.556	0.649	0.823 *	−0.364	0.854 **

** *p* < 0.01 (2-tailed), * *p* < 0.05 (2-tailed), (d and s represent dustfall and topsoil, respectively).

**Table 5 ijerph-17-02970-t005:** Principal component analysis (PCA) matrix of WSIIs and heavy metals.

Index	Rotated Component Matrix		
	RPC1	RPC2	RPC3
F^−^	0.765	0.214	0.552
Cl^−^	0.193	0.815	0.465
NO_3_^−^	0.197	0.935	0.266
NO_2_^−^	0.271	0.752	−0.236
SO_4_^2−^	0.218	0.400	0.819
Na^+^	0.064	−0.126	0.905
K^+^	0.567	0.786	−0.022
Mg^2+^	0.374	0.910	0.039
Ca^2+^	0.628	0.685	0.158
Cu	0.928	0.174	0.321
Pb	0.818	0.435	0.005
Zn	0.966	0.186	0.169
Cr	0.652	0.356	0.085
Cd	0.856	0.378	0.272
Ni	0.852	0.303	−0.269
Contribution rate %	39.93	32.39	16.32

**Table 6 ijerph-17-02970-t006:** The *I*_geo_ of heavy metals in atmospheric dustfall and topsoil in Lanzhou.

Sample	Heavy Metal	*I*_geo_ Range	Mean	Pollution Degree
Dustfall	Cu	0.5~2.3	1.2	Moderately
	Pb	1.8~2.9	2.2	From moderately to strongly
	Zn	1.6~2.5	1.9	Moderately
	Cr	−0.5~0.4	−0.1	Uncontaminated
	Cd	3.9~4.9	4.4	From strongly to extremely
	Ni	−0.7~0.2	−0.3	Uncontaminated
Topsoil	Cu	−0.5~1	0.2	Uncontaminated
	Pb	0.4~1.6	1.0	From Uncontaminated to moderately
	Zn	0~1.1	0.4	From Uncontaminated to moderately
	Cr	−0.9~0.4	−0.3	Uncontaminated
	Cd	0.9~2.2	1.6	Moderately
	Ni	−0.9~−0.4	−-0.5	Uncontaminated

**Table 7 ijerph-17-02970-t007:** The average potential ecological risks of heavy metals in atmospheric dustfall and topsoil.

Type	Metal	*E_r_^i^*		*RI*	
		Mean	Severity	Mean	Severity
Dustfall	Cu	17.5	safety	1039.9	High
	Pb	35.6	safety		
	Zn	5.5	safety		
	Cr	2.7	safety		
	Cd	972.5	High		
	Ni	6.1	safety		
Topsoil	Cu	8.7	safety	170.8	Moderate
	Pb	14.9	safety		
	Zn	2.0	safety		
	Cr	2.4	safety		
	Cd	137.5	Moderate		
	Ni	5.3	safety		

*RI* refers to the sum of potential ecological risk indexes of various elements.

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
