# Peer review of "Assessments of Water-Soluble Inorganic Ions and Heavy Metals in Atmospheric Dustfall and Topsoil in Lanzhou, China"

_ijerph, 2020, doi:10.3390/ijerph17082970_

Round 1
Reviewer 1 Report
I would suggest to combine the PCA analysis with data from ions and heavy metals. New information regarding common emission sources of these pollutants could be derived from this analysis.
The results of section 3.4 could be improved by including comparisons of dustfall and topsoil enrichment from other regions of the world. I suggest to change the title of this section by “analysis of the origin of heavy metals” or “contribution of anthropogenic heavy metals “ or something similar.
Section 3.5 Include comparisons with similar studies from other parts of the world.
It is not necessary to present Component Matrix and Rotated Component Matrix, keep only the RCM.
The conclusions section must be improved. Mentioning results already presented in the previous section should be avoided.

Reviewer 2 Report
Dear Authors and Editor,
the work presents interesting results on the chemical characteristics of atmospheric dustfall and topsoil from selected regions in China. Anyway, the paper needs to be improved following the suggestions reported in the manuscript file and in this letter. I would suggest a major revision before the publication of this paper.
Title: it is too long. I would suggest changing it, for example “Assessments of Water-soluble Inorganic Ions and Heavy Metals in Atmospheric Dustfall and Topsoil in Lanzhou, China”
Abstract: it needs to be re-write and English reviewed. See details in the pdf file
Introduction: also if the content is well structured (logical order) and Authors well describe the background data, the problem to solve and their aim, the introduction is a little bit confuse (some English corrections are necessary).
Line 40-41
WSIIs are 40 hygroscopic [6]: are you sure? Maybe the aerosol particles are hygroscopic not the WSIIs.
Materials and Methods: generally, it is well described but Authors need to add some detail:
- Please, add some geological information.
- Did you apply the quartering method?
- Did you analyse triplicate samples for WSIIs determination and acid digestion?
- Please, add the following data: reference material used for acid digestion, % recovery, precision, accuracy.
- Please, add the reference solution for ICP-MS analysis.
Figure 1: please, do not report the legend and scale on the map and add the reference for the map. The resolution is very low.
Results
Table 1 and Table 2: please check the significative figures. I would suggest adding the standard deviation (SD) for each chemical species and approximate all concentrations according to SD.
Discussion
The discussion and the interpretation of the data are too general. Author give several hypotheses for the chemical characteristics of the dustfall without any certainty. I would suggest re-elaborating it in a more critical way and adding more references to previous works, also in different areas.
Conclusions
This section is a summary non a conclusion. Please, re-write it considering a broader context, underling the utility of your work.
I am not a native English speaker, but I would suggest language editing.
Best regards,
the referee

Round 2
Reviewer 1 Report
I disagree with the authors about the purpose of PCA analysis. In pollution studies this analysis shows common emission sources of elements with high loadings in the same factor, it does not provide information about influencing factors.
In addition, combining data from heavy metals and ions in the same PCA analysis provides interesting information that is missing when running the analysis separately. For example, it shows that F- is probably released to the atmosphere by the same emission sources than most heavy metals, indicating these elements could have been released by industries. However, the authors should assess if this result has sense, according to the industries present in the study area.
Regarding point 4, RCM is the key output of principal components analysis. It has the estimates of the correlations between each of the variables and the estimated components. This matrix is derived from CM, therefore it has no sense to keep both.
Reviewer 2 Report
Dear Authors and Editor,
although some corrections were done on the revised version of the manuscript, there are still some points to solve. I report a list below.
My previous observation on the Introduction: also if the content is well structured (logical order) and Authors well describe the background data, the problem to solve and their aim, the introduction is a little bit confuse (some English corrections are necessary).
This point needs further improvements.
About my comment: Please, add some geological information, Authors added “Lines 83-87: Lanzhou is located at the Yellow River Valley and surrounded by hills with a temperate 82 continental climate. The average annual temperature is 10.3 °C. There is no extreme heat in summer 83 and no severe cold in winter. The annual average sunshine hours are 2446 hours, the frost-free 84 period is 180 days, and the average annual precipitation is 327 mm, which is mainly concentrated 85 from June to September”
I am sorry, but these are not geological characteristics.
Methods: several data are still missing
- Reference material used for acid digestion. I mean a certified solid reference material whose chemical composition is certified by several laboratories to test the acid digestion procedure.
- Reference solution for ICP-MS analysis (not internal standards). I mean certified reference solution (e.g. EnviroMAT Drinking Water - SCP Science, EP-H-3 and EP-L-3 reference solutions) whose chemical composition is certified by several laboratories to evaluate the precision and accuracy of your analysis. Although you can estimate precision by multiple analysis of the same sample, you can not estimate accuracy in this way because you do not know the “true” value.
Also, I would suggest sending the paper to some English Editing & Author Services for Research Publication.
Best regards,
The referee
